# Uncertainty Evaluation in Vision-Based Techniques for the Surface Analysis of Composite Material Components

**DOI:** 10.3390/s21144875

**Published:** 2021-07-17

**Authors:** Giulio D’Emilia, Antonella Gaspari, Emanuela Natale, Davide Ubaldi

**Affiliations:** 1Department of Industrial and Information Engineering and of Economics, University of L’Aquila, 67100 L’Aquila, Italy; EMANUELA.NATALE@UNIVAQ.IT (E.N.); davide.ubaldi@student.univaq.it (D.U.); 2Department of Mechanics, Mathematics and Management, Polytechnic of Bari, 70125 Bari, Italy; antonella.gaspari@poliba.it

**Keywords:** image analysis, angle measurement, uncertainty, composite materials, thermoforming, surface inspection

## Abstract

In this paper, a methodology is discussed concerning the measurement of yarn’s angle of two different glass-reinforced polypropylene matrix materials, widely used in the production of automotive components. The measurement method is based on a vision system and image processing techniques for edge detection. Measurements of angles enable, if accurate, both useful suggestions for process optimization to be made, and the reliable validation of the simulation results of the thermoplastic process. Therefore, uncertainty evaluation of angle measurement is a mandatory pre-requisite. If the image acquisition and processing is considered, many aspects influence the whole accuracy of the method; the most important have been identified and their effects evaluated with reference to two different materials, which present different optical-type characteristics. The influence of piece geometry has also been taken into account, carrying out measurements on flat sheets and on a semi-spherical object, which is a reference standard shape, to verify the effect of thermoforming and to tune the process parameters. Complete uncertainty in the order of a few degrees has been obtained, which is satisfactory for purposes of simulation validation and consequent process optimization. The uncertainty budget also allowed individuation of the most relevant causes of uncertainty for measurement process improvement.

## 1. Introduction

In the past few years, the use of advanced composites has become more and more popular in a wide range of activities, in particular, in the automotive and aerospace industries, due to their important characteristics, such as high strength-to-weight ratios, high stiffness-to-weight ratios, low density, wear resistance, and long fatigue life. Among advanced composites, those consisting of a thermoplastic matrix present many advantages compared to thermoset-based composites, such as chemical resistance, recyclability and the capacity to be formed and produced at a high volume rate [1].

The thermoforming process is among the most promising manufacturing processes for the production of thermoplastic-based composite parts.

However, it must be considered that during the process, several defects may be introduced, such as wrinkles, variations of thickness, residual stresses, and, in particular, deformation of the fiber textile reinforcement that influences both the structural characteristics of the product and its aesthetic characteristics [1].

In fact, fiber-reinforced plastic materials exhibit an orthotropic behavior due to fiber orientation, closely linked to the structure of the textile and to the angles between weft and warp [2,3,4].

The textile draping process is one of the most critical steps in the production of fiber-reinforced thermoplastic composite because, during the process, a flat textile has to be adapted to a 3D geometry, and this induces shear deformations and changes in the local yarn orientation, which can be critical to the performance of the material [3].

Analysis of the genesis of these defects for optimizing the manufacturing process is typically realized by performing large deformation mechanical tests, although in recent years, there have been an increasing number of studies using finite element (FE) modelling for simulating the process [5,6,7].

Simulation plays a fundamental role for process optimization, but many aspects should be taken into account when the phenomenon is so complex, with reference not only to physical and chemical characteristics of materials, but also to the real operating condition during the production process [8,9,10]. For these reasons, the integration and completion with experimental evaluations is an unavoidable aspect of the approaches aiming to optimize the production process.

To assess the conformity between measurement and modelling results, study of the causes of variability in the measurement process and their evaluation is essential [11,12,13].

Different methods are proposed in the literature for the measurement of yarn orientation in draped fabrics. For the analysis of intermediate levels, high-frequency eddy current testing is often used for the detection of fabric defects such as gaps and foreign materials [14,15,16], or for the visualization of yarns in fabrics [3]. For this method, the fibers of the composite material have to be conductive, which is the case for carbon fiber fabrics, but not, for example, for glass fiber composites.

Accurate descriptions of microstructure in terms of fiber volume and fiber orientation are possible with the analysis of 3D images of material, obtained by micro-tomography, although this technique is scarcely used and very expensive for large-scale studies [17]. Investigation of fiber orientation is also possible with scanning electron microscopy (SEM), but this kind of method cannot be easily automated [18].

Direct optical methods exist, useful for the inspection of one-layer fabrics or of the uppermost layer in multi-layer materials, based on image analysis algorithms such as edge detection or gradient methods [3,4,19], or on the analysis of the bi-directional reflectance distribution function (BRDF) [20]; an interesting and promising line of research also concerns the analysis of images acquired with polarized cameras [21,22,23].

Optical methods present various advantages, being non-destructive, contactless, relatively inexpensive and with the possibility of automation, provided they are associated with effective techniques of image analysis. They are also very versatile and applicable to different types of surfaces, if the experimental set-up is suitably developed, in particular, regarding lighting.

The measurement of yarn angles is a not trivial task; if a low level of uncertainty is required and the agreement between experimental data and the results of FE modelling is not easy to obtain, in research where a comparison is made with simulation, a deviation of 10° or more between predicted and measured fiber or shear angles is generally found [3,5,24].

In this study, a method based on a vision system for the measurement of yarn angles in glass fiber thermoplastic composites was applied to two different types of materials, and the results were compared taking into account the respective contributions of uncertainty, which can differ for various reasons: contrast between elements of the image, geometric regularity of the weaving, uniformity of the angle between weft and warp, etc.

Indications from the uncertainty budget can enable optimization of the experimental set-up and the procedure in order to obtain a suitable level of uncertainty for effective control of the production process of parts made of thermoplastic composite material. In addition, uncertainty is important information when validating simulation results, which, in the case of thermoforming processes of composite materials, presents various criticalities linked to the complexity of the phenomena and the knowledge of the actual operating conditions. This approach, based on the assessment of uncertainty, is new compared to extant work in the field.

The effect of varying the geometry of pieces is also considered, because the effect of the above aspects is modulated by the shape of pieces.

In Section 2, the materials examined are described, and the method for the yarn angle measurement is explained, together with the measurement strategy.

In Section 3, the results are presented for both materials examined, with reference to flat and 3D geometries. Moreover, an uncertainty budget is realized, for both the materials taken into account, and some considerations concerning the most relevant aspects are discussed with the aim of improving the experimental procedure.

Conclusions and suggestions for future work close the paper.

## 2. Materials and Methods

Both materials under analysis consisted of a polypropylene matrix reinforced with a woven 2-2 twill E-glass textile.

The first material considered was a commercial composite material called TRICAP^®^ (Samyang, Seoul, Republic of Korea) with a fiber volumetric content of 43.5%, and melting point of the matrix equal to 160 °C.

The second material, called TEPEX^®^ (Bond Laminates GmbH, Brilon, Germany) Roving Glass (RG) fabric, had a fiber volumetric content of 47%, and the melting point of the matrix was 163 °C.

These composite materials were fully impregnated and consolidated. All the fibers were thus sheathed with plastic, and the material did not contain any air pockets.

The two materials were very similar in mechanical properties, but they were very different in the color of the matrix, which in the first case was transparent, and in the second was black (Figure 1).

The color of the matrix is a crucial aspect for the measurement method, because it can either favor or make it difficult to identify the features of interest in the image.

The materials examined had been thermoformed at different temperatures, which has a great influence on the angles between warp and weft of the finished object.

A careful analysis of the uncertainty of measurement by the vision system was carried out, with reference to the main aspects affecting the accuracy:Surface characteristics of material (texture and optical characteristics);Surface curvature and geometry;Measurement set-up (lighting and distance of measurement);Choice of the parameters of the algorithm for image analysis (depending on the method).

### 2.1. Surface Characteristics of Material (Texture and Optical Characteristics)

The color uniformity of the first material caused greater difficulties in identifying the edges and required a more careful setting of the software parameters. To mitigate the problem, a black sheet was placed behind the composite in order to highlight the contrast between the interstices and the fabric.

On the other hand, because the matrix was transparent, the entire length of the fibers was visible, unlike the dark matrix material, where the fiber was partially submerged by the matrix itself; this was an advantage because the angle measurement could be performed on more regular and extended elements.

### 2.2. System Calibration and Surface Curvature and Geometry

Measurements were preliminarily carried out on a reference image realized by CAD software, to validate and evaluate the accuracy of the methods. Then, repeated measurements were carried out on a flat piece of the composite materials, to also evaluate the variability of the yarn angles on the planar starting plate.

In the next phase, measurements were carried out on 3D objects, with semi-spherical geometry, obtained through a thermoforming process (Figure 2). This step enabled verification of the effect of the surface geometry on the proposed approaches and validation of the simulation of the thermoforming process.

### 2.3. Measurement Set-Up

A FLIR Polarized Camera, model BFS-PGE-51S5P-C, 2448 × 2048 pixels, and 25 mm, 1:2.8 lens, were used for the acquisition of the images. The working distance of the camera from the piece was set equal to 250 mm. As for illumination, two soft-boxes with 80 W lamp bulbs were used, positioned about 800 mm apart, either side of the object.

### 2.4. Yarn Angle Measurement Methods and Parameters of Interest

The processing of the images was performed by means of high-performance software for the geometrical analysis of elements in the images: Vision Builder for Automated Inspection, by National Instruments. The algorithm for edge detection was used for the specific application; the purpose was to identify the edges of the warp and weft yarns in order to measure their relative angles.

To locate an edge, a search direction had to be set in the figure, which is a segment along which the differences of intensity between pixels are detected: edges were identified by peaks in the edge strength profile. To find a whole edge line, a region of interest (ROI) was defined, in which about 20 research lines were distributed (Figure 3), each identifying a point on the edge. Fitting of the identified points provided the searched edge. The number of research lines can affect the results; therefore, a contribution of uncertainty due to different settings is considered in Section 3.3.

The method for inspection of the composite surface involved identification of the straight edge of the individual glass yarn elements. Pairs of perpendicular and neighboring elements were selected for determining the angle θ between fibers (Figure 4).

The tow width variation throughout the fabric was considered a cause of variability.

It is necessary to point out that this kind of technique presents specific difficulties in measuring angles on curved surfaces, because the representation of a three-dimensional object on a flat image may induce a perspective error. However, if the regions considered are restricted, and the camera axis is perpendicular to the scanned region, the ROIs can be considered nearly flat, and the error is expected to be minimal.

The angle between weft and warp can be defined in two possible ways (Figure 5), considering the two supplementary angles between yarns; therefore, it was necessary to define a conventional rule. In this study, the angles were defined as those evaluated in a counterclockwise direction between weft and warp: β in Figure 5. Thus, the orientation of the piece within the image did not matter, because in each point of the piece the angle between the tows was evaluated exclusively with reference to the weft and warp orientations, regardless of external reference systems.

After a preliminary validation of the method on a reference twill fabric picture, the following experimental sequence was carried out:Measurements on a flat plate whose nominal angles are 90°:32 repeated measurements, realized with reference to the same couple of yarns. In particular, the pair was taken at random within an area of 200 × 200 mm^2^, and repeated measurements were carried out by redefining the ROIs for the identification of the edges each time;32 measurements on different couples of yarns, taken within the considered area of 200 × 200 mm^2^. Couples were considered different if at least one element of the pair was different. It was determined that the area considered was of the same extent as the flat sheet from which the semi-spherical object was obtained by thermoforming.
Measurement on the semi-spherical object. The semi-spherical surface was divided into 16 segments and 3 zones (central, intermediate and extreme), as indicated in the scheme of Figure 6.

Twelve measurements were performed on the central zone in a single image. In the intermediate and the extreme zones, twelve measurements were made for each segment on four different images.

## 3. Results

### 3.1. Measurements on the Flat Plate

The preliminary validation with respect to a reference image, where all the angles were 90°, provided satisfactory results, with the variability of repeated measurements in the order of hundredths of a degree, and the average value of the angle centered on a reference value of 90°.

The results of the tests carried out on the flat plate are summarized in Figure 7, which shows the histograms of the results obtained with reference to the same couples of yarns (Figure 7a), and to different couples of yarns (Figure 7b) for both the TRICAP and TEPEX composites.

In the first case (TRICAP), the distribution of measurements had a mean and standard deviation of 91.7° and 1.3°, respectively, when the same couple of yarns were considered in repeated measurements; a mean angle of 89.7° and standard deviation of 2.1° were obtained when different couples were considered.

For TEPEX, the distribution of measurements had a mean and standard deviation of 91.9° and 3.3°, respectively, when the same couple of yarns were considered in repeated measurements; a mean angle of 93.2° and standard deviation of 4.4° were obtained when different couples were considered.

It can be observed that the variability evaluated on the same pair of yarns considered the variability of the methods, which was affected by the way in which the regions of interest used for the measurement were selected. Instead, the variability evaluated on different couples of yarns was also affected by the variability of the measurand, which is the superficial inhomogeneity typical of the analyzed material.

### 3.2. Measurements on 3D Objects

In the present work, a semi-spherical geometric structure, which is a standard reference shape for the evaluation of draping effects, has been considered due to its simplicity in characterizing several induced phenomena, and because it is a closed geometrical shape, it most likely assists in the development of defects such as those mentioned above [6]. The 3D objects were obtained from 200 × 200 mm^2^ sheets of 0.5 mm in thickness, by means of a draping test stand, with a hemispherical punch radius of about 50 mm [6]. The TEPEX shell was thermoformed at a temperature of 160 °C, close to the melting point of the thermoplastic matrix. The TRICAP semi-sphere was obtained at a temperature of 125 °C, which was not the optimal temperature, and this could have strongly influenced the final result. The cooling rate, and therefore the crystallinity of the matrix, has been shown to not significantly affect angle measurements in separate experimental tests.

The conventional rule for angle measurements, described in Section 2.4, was applied to the case of a semi-sphere (Figure 8).

For each area described in Figure 5, the average angle (arithmetic mean over 12 measurements in each area), referred to as γ subsequently, has been calculated.

With reference to the central, intermediate and extreme zones, the average angles for each segment, in the case of TRICAP, are presented in Figure 9; for TEPEX, the results are presented in Figure 10. Notably, areas 1 and 17 represent the same segment.

In both intermediate and extreme areas, a periodic trend was observed, of greater amplitude in the most extreme areas, caused by shear deformations induced by the draping process.

The results are also presented in the heat maps in Figure 11 and Figure 12, in terms of the complement (ϑ) of angle γ:

ϑ = 90° − γ

Notably, in these two cases, the maps were rotated by 45° with respect to each other, because weft and warp were rotated by 45° with respect to the draping test stand.

The results show that in the case of the TRICAP semi-sphere, the variations with respect to the nominal angle of 90° were lower than in the case of the TEPEX workpiece. This was due to the different temperatures at which the thermoforming was carried out. In fact, in the case of TEPEX, the higher temperature (160 °C) at which the forming was realized caused the matrix to melt, which allowed the fibers to move more freely, deviating from the original orientations, compared to the case of TRICAP, in which the temperature process was set at 125 °C.

Measurements were repeated on three different pieces, both for TEPEX and TRICAP, and the results proved to be repeatable within the limits of the estimated uncertainty.

Furthermore, preliminary measurements were also carried out on semi-spheres obtained by thermoforming TEPEX at 125 °C and TRICAP at 160 °C. TEPEX and TRICAP exhibited comparable behaviors in terms of average values of angles when the thermoforming temperature was the same, even if the variability was different, as already observed in the repeatability tests.

The choice of a different temperature was to highlight the great importance of temperature in the process, from the point of view of angles between warp and weft: in future, this will be further explored.

### 3.3. Uncertainty Evaluation

The main uncertainty contributions have been identified for the measurement method, with reference to TRICAP and TEPEX materials:Image resolution is related to the camera sensor resolution and the distance from the workpiece. It can be estimated, as indicated in Figure 13, with reference to the horizontal direction, but the same considerations can be made for the vertical direction. The number of pixels in the horizontal direction, shown in Figure 13, corresponds to the length of the identified ROI for the angle measurement.The most unfavorable combinations of angle variations in vertical and horizontal directions were considered, and the results are indicated as variability in the uncertainty budget in Table 1. The contributions were different for TRICAP and TEPEX, because the elements considered for the measurements were of different lengths in the two types of materials: in the former they were longer, because the entire bundle of fibers was visible; in the second, they were partially submerged in the black matrix.The variability of the method was evaluated in the repeatability trials on the same couple of elements, with reference to TRICAP and TEPEX materials. The results, in terms of standard deviation (already discussed in Section 3.1), are reported in Table 1;The variability of the measurand was evaluated on the basis of the measurement results obtained on the same couple and different couples of elements.

In particular, we obtained the variability due to material inhomogeneity ($\;s_{m}$) from the standard deviation of measurements obtained on the same couple ($s_{\mathrm{sc}}$) and different couples ($s_{\mathrm{dc}}$), as follows:(1)$$s_{m}=\sqrt{s_{\mathrm{dc}}^{2}{-\;s}_{\mathrm{sc}}^{2}}$$
Parameter set-up: a sensitivity analysis of the main parameters of the processing algorithms was carried out, and a maximum variation of 1.5° was obtained for the angle measurement in the case of TRICAP; a maximum variation of 1° was found for TEPEX. The different variability was due to the fact that in the case of TRICAP, where the matrix was transparent and the surface appeared rather uniform, it was more difficult to determine the edges necessary for measuring the angles, and variations in the threshold settings had a greater influence on the results;Illumination: lighting conditions were varied, modifying the number of illuminators (one or two), and distances in the range 500–1500 mm with respect to the piece, keeping the other parameters constant, and the maximum variation of the angle measurement was evaluated in the two examined cases.

In Table 1, the last column (“uncertainty contribution”) was obtained by multiplying the “variability” by the corresponding “factor” to obtain the standard deviation, and then by the “sensitivity coefficient” [25]. As for the factor value, in particular, when a rectangular probability distribution could be assumed, the standard deviation was obtained by dividing the half-width of the interval by √3. When the variability was already available in terms of standard deviation, the multiplication factor was equal to one.

## 4. Discussion

As can be seen in the uncertainty budget of Table 1, the greatest contributions of uncertainty are those related to the variability of the method and of the measurand, especially regarding the TEPEX material.

In fact, TEPEX presented a greater superficial inhomogeneity, and the application of the method was more critical on this kind of material, due to the color of the matrix, which partially covered the yarns. Therefore, although the uniformity of color that characterizes TRICAP made the identification of the edges more critical (note that the parameter effect was greater in this case), the positive effect of the total visibility of the fiber bundles prevailed; thus, the edges appeared more rectilinear and longer than in the other case.

Whole standard uncertainties in the order of 2.4° for TRICAP and 4.6° for TEPEX were obtained, which was satisfactory for the purposes of validating the simulation and consequent process optimization.

## 5. Conclusions

In this paper, a method based on a vision system and image analysis for the measurement of yarn angles in glass fiber thermoplastic composites has been applied to two different types of composite materials, TRICAP and TEPEX.

The analysis showed that the thermoforming temperature strongly influences the measured angles, because the greater fluidity of the matrix allows the fibers to move more freely, causing larger variations with respect to the starting angles, nominally equal to 90°.

The materials examined were very similar from the point of view of composition, texture and mechanical properties, but they had different surface optical characteristics, in particular, very different matrix colors; this aspect is a crucial point in the measurement method, because it can favor, or, in contrast, make it difficult, to identify the features of interest in the image.

Measurements were carried out on flat sheets to determine the variability on the starting material, and then on a semi-spherical object, which is a reference standard shape, to verify the effect of thermoforming.

Furthermore, analysis of the uncertainty contributions showed that the application of the method is more critical in the case of TEPEX, because the dark matrix submerges and hides part of the fibers, making the determination of the edges less accurate.

Evaluation of the uncertainty contributions of the angle measurement enabled the determination of limits of the method, which is a mandatory result for the next phase, providing a comparison and validation of the outputs of the FE simulation, for the same semi-spherical shape, which will be the subject of future work.

## Figures and Tables

**Figure 1 sensors-21-04875-f001:**
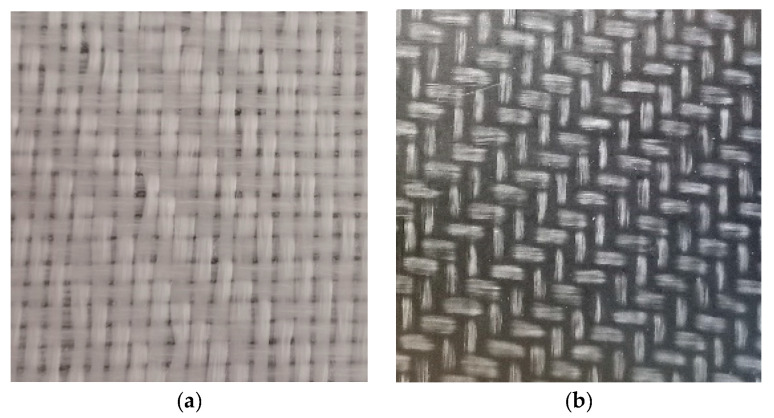
(**a**) TRICAP flat plate; (**b**) TEPEX flat plate.

**Figure 2 sensors-21-04875-f002:**
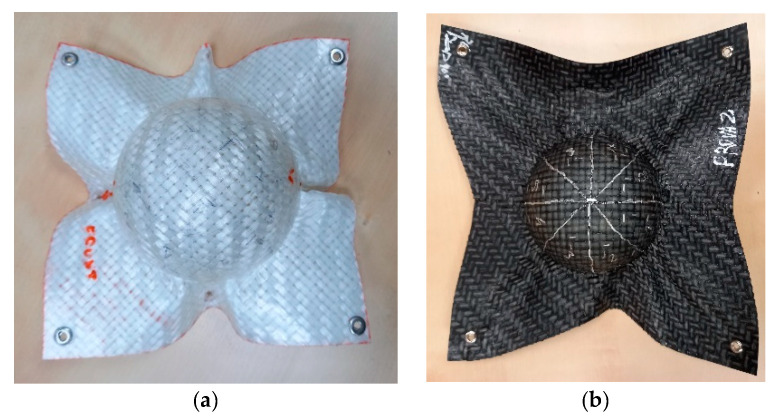
(**a**) TRICAP semi-sphere; (**b**) TEPEX semi-sphere.

**Figure 3 sensors-21-04875-f003:**
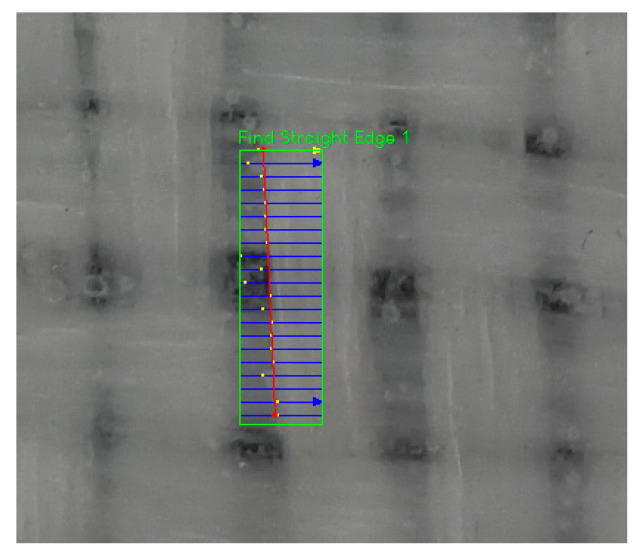
ROI for edge identification.

**Figure 4 sensors-21-04875-f004:**
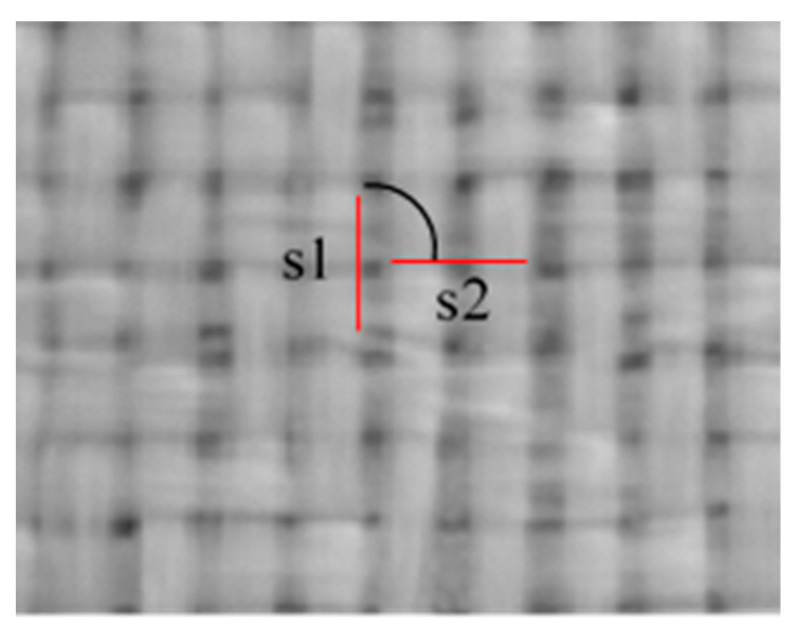
The inspection method concerning the determination of the angle between segments s1 and s2.

**Figure 5 sensors-21-04875-f005:**
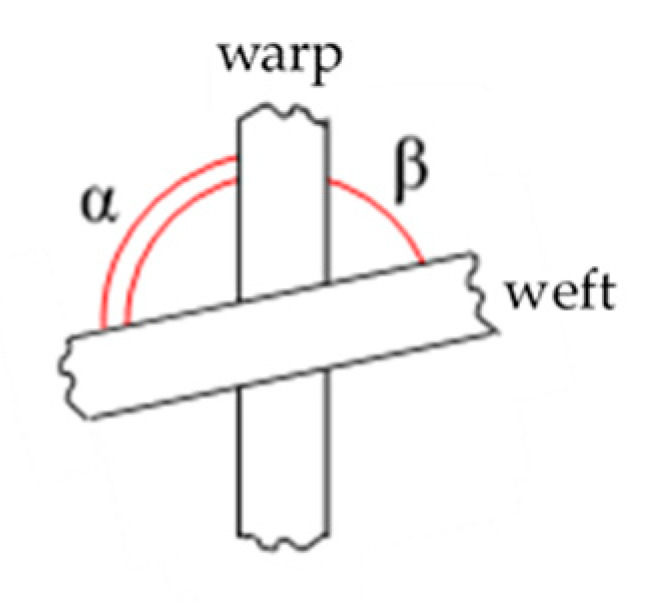
Possible definitions of the measured angles.

**Figure 6 sensors-21-04875-f006:**
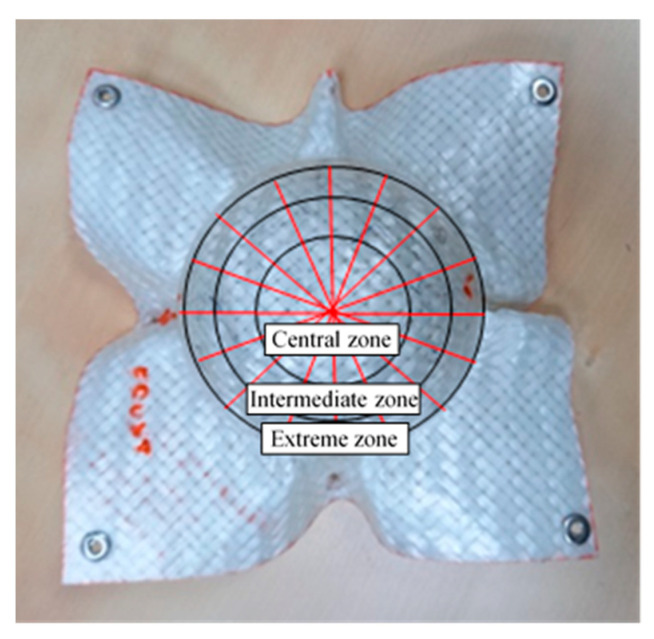
Zoning of the spherical shell.

**Figure 7 sensors-21-04875-f007:**
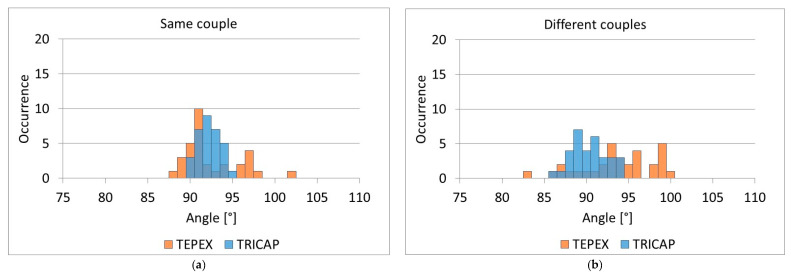
Distribution of measurements for TRICAP and TEPEX flat plates: (**a**) on the same couples of yarns; (**b**) on different couples of yarns.

**Figure 8 sensors-21-04875-f008:**
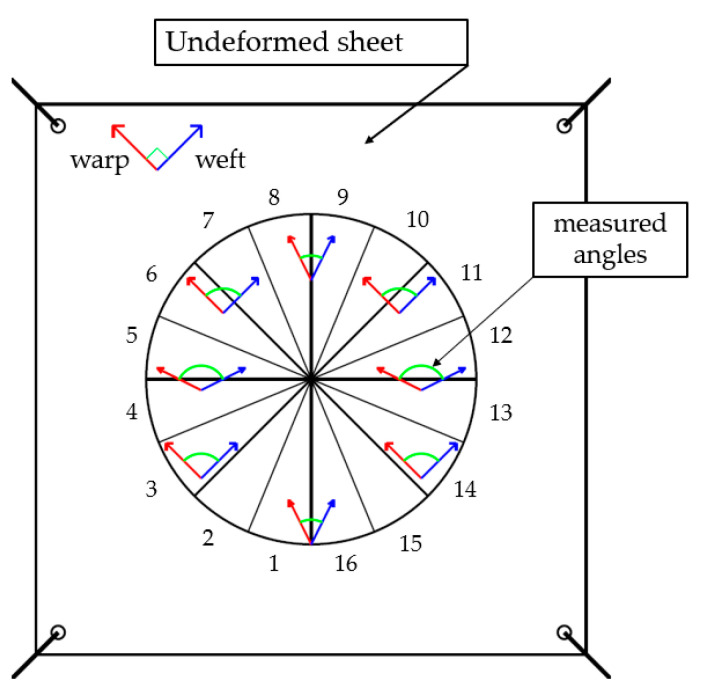
Angle definition between warp and weft on the semi-sphere.

**Figure 9 sensors-21-04875-f009:**
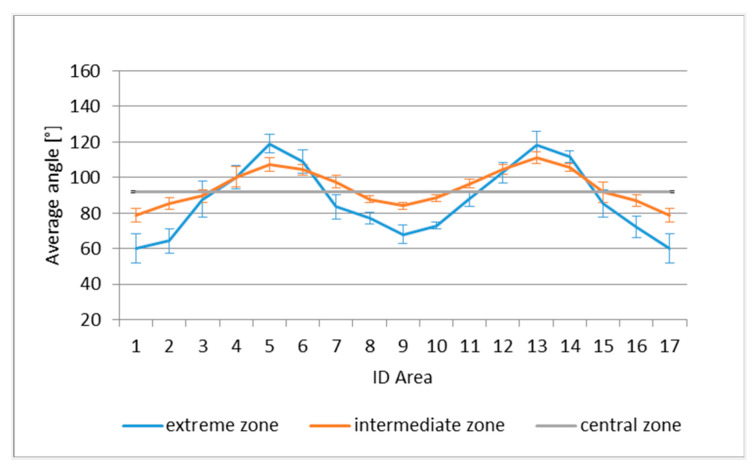
Average angles in the central, intermediate and extreme zones in TRICAP.

**Figure 10 sensors-21-04875-f010:**
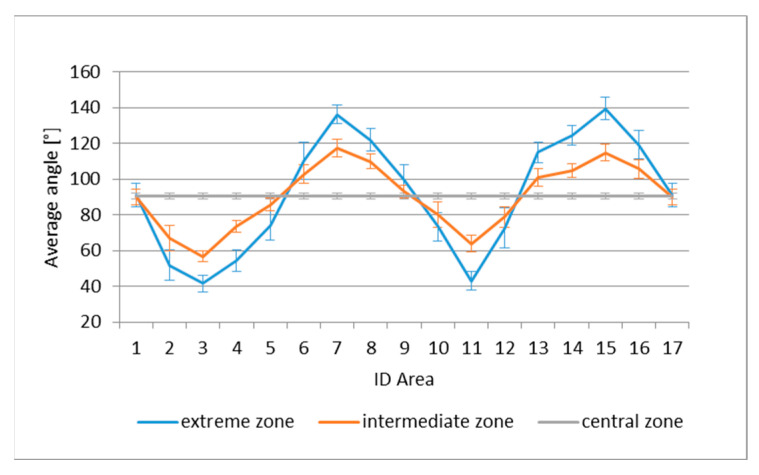
Average angles in the central, intermediate and extreme zones in TEPEX.

**Figure 11 sensors-21-04875-f011:**
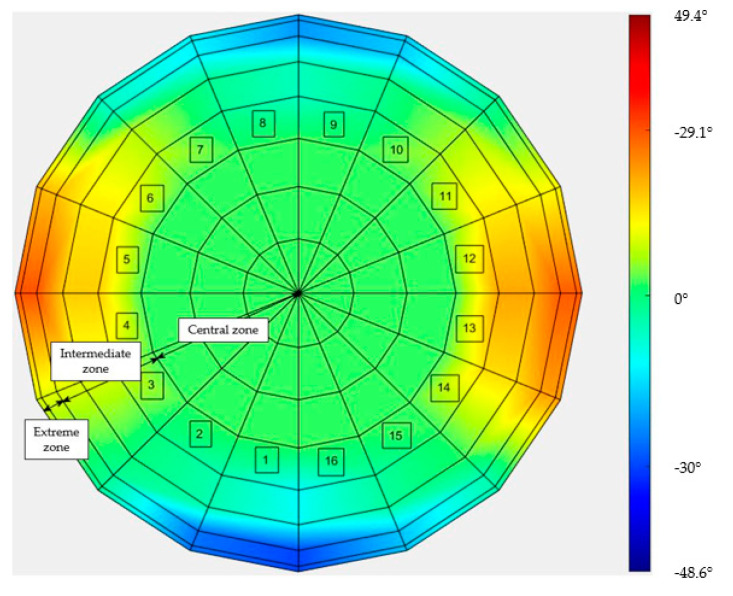
Heat map of the difference between 90° and the measured angle, in the TRICAP semi-sphere.

**Figure 12 sensors-21-04875-f012:**
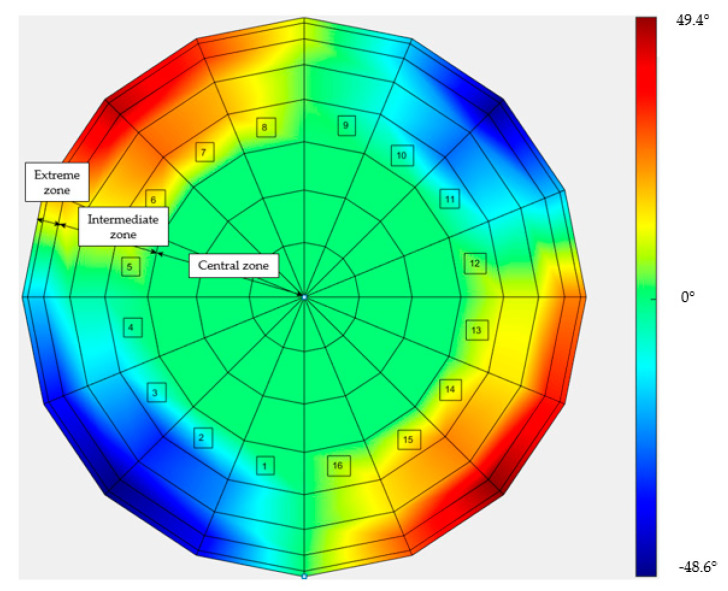
Heat map of the difference between 90° and the measured angle, in the TEPEX semi-sphere.

**Figure 13 sensors-21-04875-f013:**
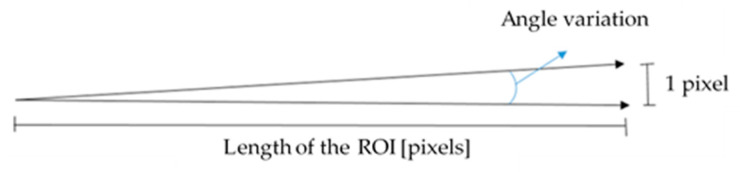
Angle variability due to the image resolution.

**Table 1 sensors-21-04875-t001:** Uncertainty budget.

	Material	Variability [°]	Distribution	Factor	Sensitivity Coefficient	Uncertainty Contribution [°]
Resolution	TRICAP	0.63	Rectangular	$\frac{1}{\sqrt{3}}$	1	0.36
	TEPEX	1.2	Rectangular	$\frac{1}{\sqrt{3}}$	1	0.69
Method	TRICAP	1.3	Gaussian	1	1	1.3
	TEPEX	3.3	Gaussian	1	1	3.3
Measurand	TRICAP	1.6	Gaussian	1	1	1.6
	TEPEX	2.9	Gaussian	1	1	2.9
Perspective	Both	1	Rectangular	$\frac{1}{\sqrt{3}}$	1	0.58
Parameter set-up	TRICAP	1.5	Rectangular	$\frac{1}{\sqrt{3}}$	1	0.87
	TEPEX	0.9	Rectangular	$\frac{1}{\sqrt{3}}$	1	0.52
Illumination conditions	Both	1	Rectangular	$\frac{1}{\sqrt{3}}$	1	0.58
Overall standard deviation	TRICAP	2.4°				
	TEPEX	4.6°				

## Data Availability

The data presented in this study are available on request from the corresponding author.

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
