# Peer review of "Uncertainty Evaluation in Vision-Based Techniques for the Surface Analysis of Composite Material Components"

_sensors, 2021, doi:10.3390/s21144875_

Round 1

Reviewer 1 Report

In the present study, vision sensing techniques and image analysis approaches were employed in an effort to assess variability of yarn angles produced due to glass fabric draping in fabrication procedures of fiber-reinforced thermoplastic composites. The image analysis results demonstrated that thermoforming temperature strongly influences the yarn angle differences since the matrix fluidity affected by an increase of temperature allows the fibers to move more freely. In addition, analysis of the uncertainty contributions showed that the application of the method is more critical in the case of TEPEX composites. A series of works carried out in the present study shows close relevance in technological advances in industries, but in research impact point of view, experimental schemes and experimental results analysis do not meet adequacy of scientific research paper.

Major comments

  1. In the experimental schemes, it can be found that there are two major variables in the experiments, which are the thermoforming temperature and the type of composite material. Although the experiment possessed the two variables, only two types of composite materials were prepared and research analysis was carried out for those. The research scheme should have prepared four different types of composites by taking the two variables into account. For example, number 1 material: TRICAP formed in temperature 120Ëš, number 2 material: TRICAP formed in temperature 160Ëš, number 3 material: TEPEX formed in temperature 120 Ëš, and number 4 material: TEPEX formed in temperature 160 Ëš.
  2. It seems that the image analysis was carried out for only one sample of TRICAP and TEPEX, respectively. The image analysis results could be more reliable if the image analysis had been carried out for replicated samples.
  3. Originality of the present study should be reinforced.

Minor comments

  1. In ‘section 2. Materials and Methods’, some details of composites fabrication procedures needs to be addressed. For example, what is the fabrication procedure? VARTM or VIP?
  2. In ‘section 3.1 measurement on the flat plate’, it is recommended to provide additional explanations on meaning of ’the same couple of yarns’ and ‘the different couple of yarns’
  3. In ‘section 3.2 Measurement on 3D objects’, line 242-243, it was stated that ‘the angles are defined as those evaluated in a clockwise direction…’ It seems that it is not a clockwise direction but a counterclockwise direction.
  4. In Figure 8 and 9, the format of graphs was not consistent with that of Figure 6.
  5. In Figure 10 and 11, Heat map of composite TEPEX was only found. It is recommended to provide the heat maps of composite TRICAP.
  6. In Figure 12, the text font was not consistent.

Reviewer 2 Report

An interesting article, but mostly of the practical importance. The research analysis supported by an appropriate literature review.

Critical comments to the article:

  1. Drawings should be placed where they are referred to in the text, e.g.: Fig. 1, 2.
  2. How to interpret: "...a certain number of research lines are distributed (Figure 3)." - (p.4, section 2.4)? How many? Under these conditions, does it matter? 
  3. Why the TRICAP semi-sphere has been obtained at
    a temperature of 125° (p.7, section 3.2)? The authors themselves admit that this is not the optimal temperature.
  4. Figure 10 has the wrong description.
  5. The heat-maps are similar, and the course of the average angles in Figures 8 and 9 is different.
  6. Are ID Area 1 and 17 the same (Figures 8 and 9)?
  7. In section 3.3, the main components of uncertainty have been identified. They are not analyzed in the article, e.g.: ilumination, image resolutions. There is no analysis - some description.

Reviewer 3 Report

Abstract should state that the studied yarns are glass fibre.

Line 105: Tepex (RG: roving glass?) fibre type and weave style (2x2 twill?)?

Line 117: increasing PP crystallinity will generate more opaque matrix!

Line 152: FLIR = thermal imaging camera?

Line 153: camera lens, aperture and focal depth?

Line 204: how do the authors ensure the fabric edge and image edge are parallel?.

How do the authors ensure no shearing of the fabric when positioned relative to the camera?

Have the authors only considered that tows edges run parallel throughout the fabric with no tow width variation?

Line 236: rate of cooling will affect crystallinity and hence opacity?

Line 247: How is the average angle γ calculated from α and β (arithmetic or geometric mean?).

Figures 8 and 9 would benefit from inclusion of both warp and weft data?

Line 268: fusion is a synonym for melting, or enhanced fibre/matrix bonding?

Figures 10 and 11 are superficially identical?

Lines 277 and column 7 of Table 1: how is the uncertainty contribution calculated?

Line 279:  “[uncertainty contribution] is related to the camera sensor resolution and the distance from the workpiece” but the tests were all conducted with single values for both these parameters?

Line 312: now do lighting conditions vary from that indicated at line 154?

Line 340: but the processing temperatures are also very different?

Round 2

Reviewer 1 Report

The author's responses to the review comments have been examined.

The authors addressed regarding most of comments but following point should be further addressed.

The response to Major comment 1.

Author wrote "Furthermore, preliminary tests have carried out by inverting the thermoforming temperatures considered for TEPEX and TRICAP, and similar values have been found for the two materials, in terms of average values of angles, at the same temperature, even if the variability is different, as already observed in the repeatability tests."

In the response, it was not clear. In the first manuscript, it was said that thermoforming temperature greatly affected the average value of angles of fabric. However, in the author's response, it was said "inverting the thermoforming temperatures considered for TEPEX and TRICAP, and similar values have been found for the two materials, in terms of average values of angles" Please eliminate confusions and enhance clarity.

The response Minor comment 4.

The format of Figure 9 and 10 in the new manuscript is not still consistent with that of the Figure 7. Please check border line of the graph, border line of image itself, and style of the legend.

In addition, it is recommended to address the corrections together with the authors's response since it was hard to find corrections made for corrresponding reviewer's comments in the new manuscript.
